# NEURO-COGNITIVE RADIOS FOR DYNAMIC SPECTRUM ACCESS

## ABSTRACT

Neuromorphic computing is an emerging brain-inspired information processing paradigm that is well-suited for energy-efficient, real-time, and adaptive applications such as Dynamic Spectrum Access (DSA). In this paper, we develop and present the Neuro-Cognitive Radio (NCR) framework—a neuromorphic-based learning architecture to address challenging decentralized DSA scenarios where multiple source-destination pairs share a limited number of spectrum bands. NCR combines spiking neural networks (SNNs) and reinforcement learning (RL) architectures, allowing sources to adapt their transmission strategy over time to maximize their own throughput while promoting fairness across the entire wireless network. We evaluate NCR in several network settings, including settings with time-varying number of available spectrum bands, and compare with an equivalent Deep Q-Network (DQN) architecture that uses traditional multi-layer perceptron (MLP). Our simulation results show that NCR consistently achieves higher fairness than DQN, while keeping similar throughput levels. This work constitutes a promising initial step toward neuromorphic-based solutions for DSA.

## 1 INTRODUCTION

The importance of Dynamic Spectrum Access (DSA) for beyond-5G wireless networks has been highlighted in the U.S. National Spectrum Strategy (National Telecommunications and Information Administration, 2023), in the OSTP National Spectrum Research and Development Plan (National Science and Technology Council, 2024), in the ITU Report (International Telecommunication Union, 2021), and in additional recent reports. DSA (Zhao & Sadler, 2007; Akyildiz et al., 2006) is the process by which sources (*i.e.*, wireless devices) autonomously and adaptively select spectrum bands for communication, based on real-time observations of spectrum availability and usage, without relying on fixed spectrum assignments. A key challenge of DSA is interference, also called collision, which may happen when two or more sources transmit information at the same time using the same band. Destinations often cannot decode the information when a collision occurs, resulting in a waste of network resources. In traditional centralized wireless networks, *e.g.*, 5G cellular, the cell tower coordinates transmissions among different sources to avoid collisions (Qamar et al., 2019).

In this paper, we focus on *decentralized DSA wireless networks* (Naparstek & Cohen, 2019; Yu et al., 2019; Zhang et al., 2025) in which sources cannot coordinate transmissions by sharing information with each other (explicitly or implicitly). Each source independently learns to adjust its transmission strategy—specifically, which spectrum band to use—over time. The only feedback available to each source is the result of its own transmission attempts (*i.e.*, success or collision). Sources have no prior knowledge of the total number of source-destination pairs in the network or their transmission strategies. The objective of each source is to optimize its own throughput while promoting fairness across the network. The ideal network should have no collisions and all sources with identical throughput levels. Transmission strategies for each source in decentralized DSA wireless networks should be efficient, adaptive, fair, and deployable on resource-constrained radios.

In recent years, Reinforcement Learning (RL) has gained attention as an effective method for facilitating DSA in decentralized wireless networks. Through interactions with the network, RL enables individual sources (also called agents) to learn and adapt their transmission strategies over time. However, most RL-based solutions proposed in the literature, as a recent survey in (Guimarães et al.,

2024), use complex architectures—*e.g.*, Dueling Deep Q Networks with Likelihood Hysteretic Implicit Quantile Network (Zhang et al., 2025)—complex training procedures (*e.g.*, centralized training and decentralized execution Lu et al. (2022); Chang et al. (2023); Naparstek & Cohen (2019); Yu et al. (2019)), and/or rely on the capability of sources to coordinate transmissions by sharing information with other sources implicitly or explicitly (Sohaib et al., 2022; Xu et al., 2020; Bokobza et al., 2023; Janiar & Pourahmadi, 2021), thereby limiting the applicability of these RL-based solutions to real-world DSA scenarios with resource-constrained radios.

Neuromorphic computing (NMC) is an emerging information processing paradigm inspired by the structure and function of the human brain, mimicking its sparse and event-driven communication, where neurons fire only when necessary, as described in section 3.1. NMC has been successfully applied to numerous problems (Schuman et al., 2022; Eshraghian, 2023; Parpart et al., 2023) that require energy-efficient, real-time adaptation to dynamic scenarios including: (i) drone navigation and perception under tight energy budgets (Chowdhury et al., 2025); (ii) visual SLAM using "event cameras" that capture scenes asynchronously, consuming less power (Tenzin et al., 2024); and (iii) power-efficient sensing using micro-Doppler radars (Intel Corporation, 2025).

We believe that these inherent features of NMC, namely real-time adaptation and energy-efficiency, make it a promising approach to address challenging DSA scenarios. **To the best of our knowledge, this is the first paper to bring neuromorphic methods to DSA.** We propose Neuro-Cognitive Radio (NCR): a spiking neural network (SNN)-based RL architecture that learns to adapt its transmission strategy over time, aiming to maximize its own throughput while striving for network-wide fairness. NCR's training and execution are fully decentralized and do not rely on information sharing among sources for coordinating transmissions.

We use simulations to evaluate NCR in $50+$ network settings (static and time-varying) and compare NCR with an equivalent Deep Q-Network (DQN) architecture that uses traditional multi-layer perceptron (MLP). Our simulation results show that NCR can increase fairness—as measured by Jain's fairness index (Jain et al., 1984)—by a factor of $2.5$ when compared with DQN, while keeping comparable throughput levels. While the results in this work showcase a NMC solution that is capable of achieving competitive and reliable results across several DSA network settings, exploring potential gains of NMC implementations in terms of energy-efficiency is left for future work. We believe that the results in this paper are a promising initial step toward neuromorphic-based solutions for DSA.

Over the next sections, we formally describe the DSA problem (section 2), explain our NCR solution in greater detail (section 3), discuss our simulation experiments (section 4), and provide some final remarks on how to leverage and further expand our model's capabilities (section 5).

## 2  DYNAMIC SPECTRUM ACCESS IN DECENTRALIZED WIRELESS NETWORKS

We consider a decentralized wireless communication network with $M$ source-destination pairs transmitting packets via $N \geq 2$ orthogonal frequency bands. We assume that sources always have packets to transmit and that destinations are constantly listening to all bands. In each time slot $t \in \{1, ..., \mathcal{T}\}$, each source $m \in \{1, ..., M\}$ takes an action $a_m(t) \in \{0, 1, ..., N\}$, where $a_m(t) = 0$ means that source $m$ is idle, and $a_m(t) = n \geq 1$ represents a transmission in band $n$ during time slot $t$.

We then define $o_m(t)$ as the outcome of source $m's$ action in time slot $t$. There are three possibilities for $o_m(t)$: if source $m$ idles during $t$, then $o_m(t) = 0$; if only source $m$ transmits in the selected band, then the transmission is successful and $o_m(t) = 1$; finally, $o_m(t) = -1$ when two or more sources transmit in the same band, leading to a packet collision.

Thus, in this DSA problem, sources share no information to coordinate transmissions. At any given time slot $t$, source $m$ knows only about its current and previous actions and outcomes—$\{a_m(k)\}_{k \leq t}$ and $\{o_m(k)\}_{k \leq t}$, respectively. Furthermore, sources have no prior knowledge about the network topology or size $N$. This problem is analyzed across different network settings, *i.e.*, with different numbers of agents and bands.

In addition to this baseline problem, this work also considers time-varying conditions by adding a jammer—a wireless device that intentionally emits radio signals to disrupt or interfere with legitimate transmissions, reducing the available spectrum. We then repeat the same network settings as

the baseline problem under a jamming episode, where the jammer enters a band at a specific instant, holds the band continuously, and leaves it.

# 3 PROPOSED SOLUTION: NEURO-COGNITIVE RADIO (NCR)

We present some fundamental concepts in neuromorphic computing followed by a detailed explanation on how to apply such ideas to our specific DSA problem.

## 3.1 FUNDAMENTALS OF NEUROMORPHIC COMPUTING

Drawing inspiration from the human brain, neuromorphic computing is fundamentally different from its digital (von Neumann) counterpart on many levels. Apart from spike versus binary data representation, other major differences include operation (parallel or sequential) and organization (co-located or separated memory and processing units) (Schuman et al., 2022; Eshraghian, 2023). Moreover, while digital processors are time-driven (synchronous), neuromorphic processors are event-driven (asynchronous)—a major feature that has been leveraged to build energy-efficient computing platforms (Eshraghian, 2023).

The building block of neuromorphic computing is the neuron. There are multiple models that accurately describe the neuron. Some optimize for higher biological fidelity (Hodgkin-Huxley, Morris-Lecar) whereas others focus on computational efficiency (Izhikevich, AdEx IF). Our work uses the leaky-integrate-and-fire (LIF) neuron due to its simplicity (Hunsberger & Eliasmith, 2015).

The LIF model approximates the neuron's membrane potential $U(t)$ to a low-pass filter circuit made of a resistor $R$ and a capacitor $C$—which is valid from a biological standpoint—evolving over time as

$$\tau \frac{dU(t)}{dt} = -U(t) + RI(t), \tag{1}$$

where $\tau = RC$ is the circuit's time constant, and $I(t)$ is the current flowing through the circuit at any given time $t$. If the current $I(t)$ is constant over time, we obtain

$$U(t) = RI + (U_0 - RI)e^{-t/\tau}, \tag{2}$$

where $U_0$ is the membrane potential at $t = 0$. Defining the decay rate as $\beta = e^{-1/\tau}$, we can rewrite $U(t)$ in a discrete time domain via the forward Euler method as follows

$$U[t] = \beta U[t-1] + (1-\beta)I[t] \tag{3}$$

From a ML perspective, it is useful to write $I[t]$ as $WX[t]$, where $W$ is the weight matrix and $X[t]$ is a vectorized input decoupled from effects of $\beta$. By doing so, we can write

$$U[t] = \beta U[t-1] + WX[t] + S[t-1]\theta, \tag{4}$$

which has three terms: decay ($\beta U[t-1]$), input ($WX[t]$), and reset ($S[t-1]\theta$), in which $S[t] = 1$ if $U[t] > \theta$, and $S[t] = 0$ otherwise.

A practical way to understand $U[t]$ is that the membrane potential decays over time according to its $\beta$ factor and increases whenever a spike arrives at the neuron. At any time slot $t$, if the membrane potential $U[t]$ reaches its threshold value $\theta$, the neuron fires a spike to its neighboring neurons and its potential resets to zero. In our model, both $\beta$ and $\theta$ are tunable hyperparemeters—to be discussed in greater detail in section 3.2.

## 3.2 NCR: STATES, SNN ARCHITECTURE, TRAINING PIPELINE, AND REWARDS

In this section, we describe our proposed solution to the DSA problem, dubbed Neuro-Cognitive Radio (NCR). Each source is a NCR running a neuromorphic agent. In time slot $t$, the agent's state ($s_m(t)$) includes its own actions, outcomes, and binary time references from the previous $T$ time slots. The binary time references (Zhang et al., 2025) are represented in modulo 16, *i.e.*, mod(t,16), using 4 bits, allowing agents to find transmission patterns of different lengths by adaptively ignoring bits. Mathematically, for each $k \in \{t - T, \ldots, t - 1\}$, the state $s_m(t)$ will include

$$[\text{time-ref}(k), \text{one-hot}(a_m(k)), o_m(k)] \in \mathbb{Z}^F, \tag{5}$$

where the feature size $F$ is given by $F = 4 + A + 1$, and $A$ is the action space size, *i.e.*, $A = N + 1$ with all frequency bands and idle. Aggregating equation 5 for $k \in \{t - T, \dots, t - 1\}$, we have $s_m(t) \in \mathbb{R}^{T \times F}$ which is then converted into a torch tensor with dimension $1 \times T \times F$ to facilitate our model training pipeline.

Now, we describe the architecture of the SNN, which is composed of an input layer, one hidden layer, and an output layer. For each time index $k \in \{t - T + 1, \dots, t\}$, the input projection is done through a hidden linear layer (size $H$) that gives us $z_k = W_{in}s_m(k) + b_{in} \in \mathbb{R}^H$, where $W_{in}$ and $b_{in}$ are the input weight matrix and bias vector, respectively. Then $z_k$ acts as input to a series of "spiking blocks". Each block is made of a linear layer (size $H$) followed by a LIF layer—analogously to the traditional structure of a multi-layer perceptron. The dimensions of the SNN architecture are $(input_{dim}, hidden_{dim}, output_{dim}) = (F, F, N + 1)$.

Now, we explain how the learning process works through the spiking backbone. Let $L$ be the total amount of spiking blocks and let's initialize the first block ($l = 1$) input embedding as $h_{0,k} = z_k$. For each block $l \in \{1, \dots, L\}$, we have the following forward pass mechanism:

1. Linear: $a_{l,k} = W_l h_{l-1,k} + b_l$;

2. Leaky integrate: $\tilde{U}_{l,k} = \beta_l U_{l,k-1} + (1 - \beta_l)a_{l,k}$;

3. Threshold: $S_{l,k} = \mathcal{H}(\tilde{U}_{l,k} - \theta_l)$;

4. Reset: $U_{l,k} = \tilde{U}_{l,k} - \theta_l S_{l,k}$; and

5. Spike output: $h_{l,k} = S_{l,k} \in \{0,1\}^H$,

where the subscript $l, k$ stands for the spiking block $l$ at a given time index $k$, $\mathcal{H}$ is the Heaviside function, and $U, \beta, \theta, S$ have the same definitions as those from section 3.1. The forward pass output is the last step in the last layer, *i.e.*, $z = S_{L,T} \in \{0,1\}^H$, which remains in a spiking format. Then, to map it back to the problem's original output space (*i.e.*, the number of bands), we use the Q-value concept—a way to represent our action quality at a given time slot $k$. We do so via another linear layer: $Q(s,a) = W_{out}z + b_{out} \in \mathbb{R}^A$, where $A = N + 1$.

To select an action, the agent turns a decision score into a choice through the $\varepsilon$-greedy policy, by taking $\operatorname{argmax}_a Q(s,a)$ with probability $1 - \varepsilon$—aiming to balance the exploitation of promising bands and the exploration of new ones. Then, for each action, we save sequences $(s, a, r, s', d)$ of length $T$, where $r$ is the immediate reward (to be defined later in this section) received after taking action $a$, $s'$ is the updated state, and $d$ is the "done" flag. After saving the $(s, a, r, s', d)$ sequences, and sampling a batch with size $(B, T, F)$, we define the target ($y$) and the loss ($\mathcal{L}$) functions as

$$\begin{cases} y = r + \gamma(1 - d)\max_{a'}Q(s', a') \\ \mathcal{L} = (y - Q_{\text{online}(s,a)})^2 \end{cases}. \tag{6}$$

Backpropagation is the next step. As previously described, the forward pass uses the Heaviside function, which is not differentiable. A common way to address this issue is to differentiate a function with similar format—also known as the surrogate gradient approach. Mathematically,

$$dS_t/d\tilde{U}_t \approx g_\zeta(\tilde{U}_t - \theta), \tag{7}$$

in which $g_\gamma$ is the surrogate gradient. In practical terms, this means that we keep the step function $\mathcal{H}$ in the forward pass, but we use an approximation to it in the backpropagation step. Common $g_\zeta$ alternatives include sigmoid, fast sigmoid, and arctan (Eshraghian, 2023), given respectively by:

$$\begin{cases} g_\zeta(u) = (1/\zeta)\sigma(u/\zeta)[1 - \sigma(u/\zeta)] \\ g_\zeta(u) = (1 + |u|/\zeta)^{-2} \\ g_\zeta(u) = (1/\zeta)(1 + (u/\zeta)^2)^{-1} \end{cases}, \tag{8}$$

where $\sigma(u) = 1/(1 + e^{-u})$, and $\zeta$ determines the "shape" of the surrogate function—sharper or smoother. Finally, we use Adam optimizer, update the hyperparameters, and periodically copy $Q_{online}$ to the actual target. Having completed a cycle, we update our states $s \leftarrow s'$, and continue to the next time slot.

The reward of agent $m$ at the end of time slot $t$ is given by

$$r_m(t) = \begin{cases} 0.15 \times (1.5 - w_m(t)) \text{, if } o_m(t) = 1, \text{ [successful transmission]} \\ -1 \times w_m(t) \text{, if } o_m(t) = -1, \text{ [collision between two or more agents]} \\ -0.6 \text{, if } o_m(t) = 0 \text{ and } \sum_{k=t-L}^{t} a_m(k) = 0, \text{ [idle for more than } L \text{ slots]} \\ 0.03 \text{, otherwise} \end{cases} \quad (9)$$

with $w_m(t)$ representing the weight associated with agent $m$ during time slot $t$, where

$$w_m(t) = 0.4 W_1 + 0.6 W_2 / (N - 1) \text{, and} \quad (10)$$

$$W_1 = \sum_{k=t-L}^{t-1} \mathbb{I}_{\{a_m(k)=a_m(t)\}} (2^{k-t} |o_m(k)|) \text{,} \quad (11)$$

$$W_2 = \sum_{n=0}^{N} \mathbb{I}_{\{n \neq a_m(t)\}} \sum_{k=t-L}^{t-1} \mathbb{I}_{\{a_m(k)=n\}} (2^{k-t} |o_m(k)|) \text{.} \quad (12)$$

The value of $W_1$ increases with the number of transmissions in the recent past $k \in \{t-L, \ldots, t-1\}$ using band $a_m(t)$. The value of $W_2$ increases with the number of transmissions in the recent past $k \in \{t-L, \ldots, t-1\}$ using bands other than $a_m(t)$. The multiplicative factor $2^{k-t}$ gives more importance to recent events. A high weight $w_m(t)$ reduces the reward of a successful transmission and increases the collision penalty.

As for the actual NCR implementation, all experiments have been conducted in Python. The SNN has been implemented via the snnTorch package (Eshraghian, 2023), which requires the latest versions of some additional packages (*e.g.*, numpy, pandas), as well as nir (above 1.0.6) and nirtorch (above 2.0.5).

## 4 SIMULATION RESULTS: COMPARING SNN AND DQN

We evaluate the performance of the proposed Neuro-Cognitive Radio solution, which utilizes a spiking neural network (SNN) backbone, against a baseline deep Q-network (DQN) approach (similar to Naparstek & Cohen (2019); Yu et al. (2019)) across several distinct network settings with different numbers of agents $M$ and bands $N$. The baseline DQN architecture is identical to the SNN architecture described in section 3.2, with the same dimensions $(input_{dim}, hidden_{dim}, output_{dim}) = (F, F, N + 1)$, but using multi-layer perceptrons instead of spiking neurons. All experiments are conducted over a time interval sufficient to achieve full convergence within the specified time slots. All experiments use the same hyperparameters, described in Table 1. Performance is assessed using two key metrics: (i) Jain's fairness index (Jain et al., 1984), which measures how equitably the spectrum is shared among agents (with 1 indicating perfect fairness); and (ii) total throughput, defined as the moving average number of successful transmissions per time slot (windowed by the last 500 time slots) normalized by the number of bands $N$—reflecting overall spectrum utilization efficiency.

Table 2 compares the performance of NCR and DQN in six network settings. In all settings, the SNN-based NCR achieves substantially higher fairness indices compared to DQN, while maintaining nearly equivalent total throughput. For instance, for $M = 9$ agents and $N = 2$ bands, fairness improves by a factor of 2.5, increasing from 0.3446 (DQN) to 0.8672 (NCR), with throughput remaining at 0.95. This indicates that our SNN architecture enables more equitable access without sacrificing efficiency, driving the network's agents to avoid collisions. Similar trends hold for $M = 7$ and $N = 5$ (fairness: 0.8041 to 0.9732) and $M = 10$ and $N = 8$ (fairness: 0.8670 to 0.9706), where throughput reduces marginally from 0.9860 to 0.9688 and 0.9843 to 0.9685, respectively. **The results in Table 2 highlight the SNN's ability to promote fair spectrum sharing in decentralized networks through its event-driven processing of time-dependent transmission patterns.**

To further illustrate the results in Table 2, Figure 1 depicts the evolution of individual agent success rates (throughput per agent), along with aggregate collisions per agent and idle rates per band, over time. Rates are computed as moving averages over 500 time steps for smoothness. In the plots on the left side of Figure 1, across all network settings, SNN agent success rates exhibit initial fluctuations during exploration but rapidly converge to similar values, **demonstrating the SNN's ability to find suitable transmission patterns for every agent, leading to superior fairness results.** For $M = 9$ agents and $N = 2$ bands (Figure 1a), all nine agents stabilize around a success rate of approximately 0.2, with collisions dropping sharply from 0.5 to near 0 and idle rates remaining low. For $M = 7$ and $N = 5$ (Figure 1c), all agents converge to rates around 0.15, reflecting fair distribution of the 5

Table 1: Hyperparameters used in all experiments.

| Parameter | Value |
|---|---|
| LIF neuron decay factor ($\beta_0$) | 0.25 |
| LIF neuron threshold ($\theta_0$) | 0.03 |
| Discount factor ($\gamma$) | 0.9 |
| Exploration rate ($\epsilon$) | $5 \times 10^{-2}$ |
| Exploration rate decay ($\epsilon_{\text{decay}}$) | $6.5 \times 10^{-6}$ |
| Minimum exploration rate ($\epsilon_{\text{min}}$) | $8 \times 10^{-3}$ |
| Learning rate ($\tilde{\mu}$) | $5 \times 10^{-4}$ |
| Buffer size | 500 |
| Target network's replace rate | 50 |
| Batch size ($B$) | 64 |
| Reward history length ($L$) | 16 |
| Temporal length ($T$) | 15 |
| Number of hidden layers | 1 |
| Surrogate gradient function | arctangent |

Table 2: Fairness and throughput achieved by DQN and NCR across multiple network settings (experiments).

| #Agents M | #Bands N | DQN | | NCR | |
|---|---|---|---|---|---|
| | | Fairness | Throughput | Fairness | Throughput |
| 10 | 8 | 0.8670 | 0.9843 | 0.9706 | 0.9685 |
| 9 | 3 | 0.6393 | 0.9527 | 0.8929 | 0.9520 |
| 9 | 2 | 0.3446 | 0.8960 | 0.8672 | 0.9160 |
| 7 | 5 | 0.8041 | 0.9860 | 0.9732 | 0.9688 |
| 4 | 2 | 0.6651 | 0.9060 | 0.7721 | 0.9830 |
| 3 | 2 | 0.8723 | 0.9860 | 0.8864 | 0.9760 |

bands. For $M = 10$ and $N = 8$ (Figure 1e), all agents settle at 0.097 each, with minimal variance. Collision and idle rates decrease steadily, contributing to high throughput.

In contrast, in the plots on the right side of Figure 1, DQN agents show greater disparity in their throughput, explaining the lower fairness indices in Table 2. For $M = 9$ and $N = 2$ (Figure 1b), success rates diverge, with a single agent dominating at 0.8 while others languishing below 0.4, and agents 2, 3, and 7 remaining (almost) silent. For $M = 7$ and $N = 5$ (Figure 1d), although convergence is better than NCR, noticeable throughput gaps remain with rates ranging from 0.1 to 1. Similarly, for $M = 10$ and $N = 8$ (Figure 1f), the throughput spread is noticeably larger than in SNN with rates ranging from .2 to 1. **These plots underscore Neuro-Cognitive Radio's superiority in achieving balanced, adaptive spectrum access through neuromorphic principles.**

We highlight that the only difference between NCR and DQN is that the first utilizes spiking neurons while the second utilizes multi-layer perceptrons. Taken together, the results in Table 2 and Figure 1 suggest that a neuromorphic (spiking) backbone is effective for fair, decentralized DSA: it achieves a favorable fairness–throughput tradeoff while producing stable per-agent behavior.

To further showcase NCR capabilities, we run several additional experiments, with $M \geq 2$ agents and $N \geq 2$ bands, as illustrated in Figure 2. Across all the 45 network settings, all using the same hyperparameters from Table 1, we can observe a high Jain's fairness index: greater than 0.95 in more than 50% of the experiments and achieving its lowest value (0.8670) at 9 agents and 2 bands, which is the setting illustrated in Figure 1a. Moreover, we also observe high throughput, with 42 (out of 45) experiments having throughput greater than or equal to 0.95.

To assess how well NCR responds to time-varying conditions, we examine network settings in which a jammer disrupts transmissions by occupying a single spectrum band for a period of time. The

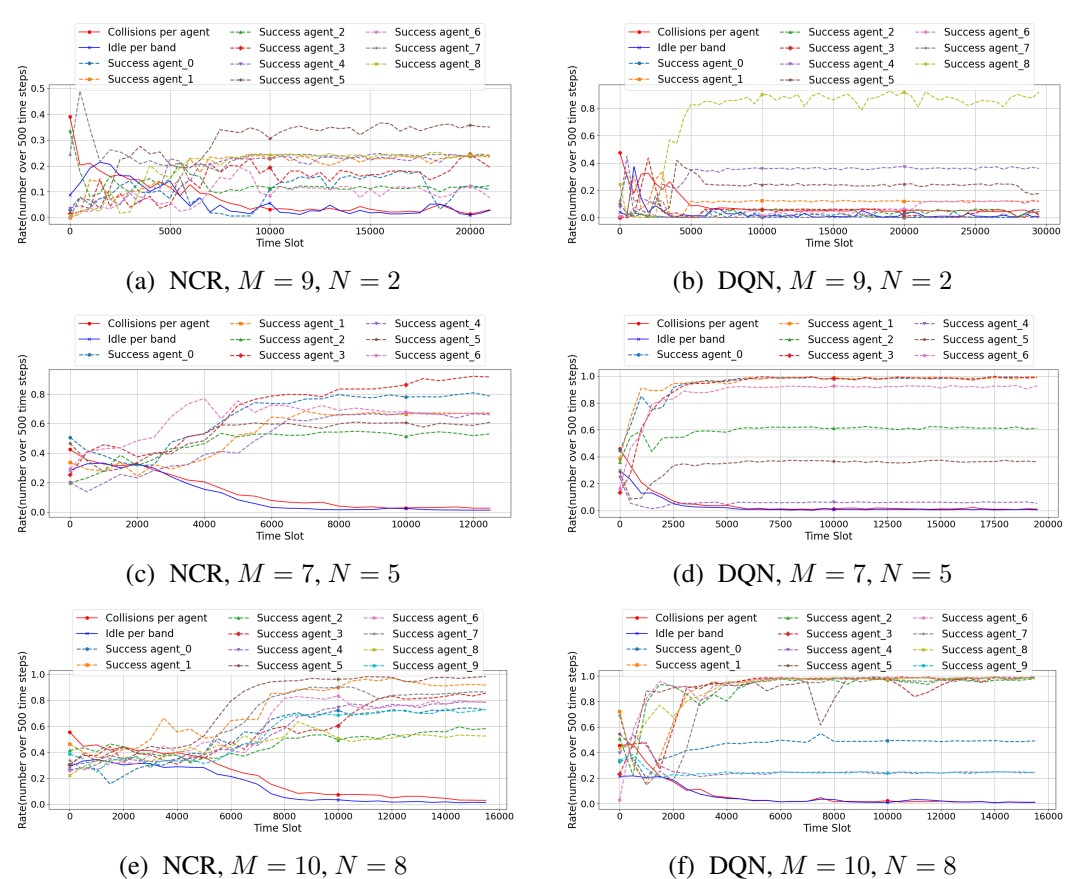

Figure 1: Idle rate, collision rate, and per-agent throughput over time $t$. Left column: Neuro-Cognitive Radio (NCR). Right column: DQN. Panels (a)–(f) correspond to the three $(M, N)$ settings.

spectrum band chosen by the jammer and the jamming period are unknown by the agents. In Figure 3, the jammer occupies the last band $N$ between time slots 20,000 and 30,000, and between time slots 40,000 and 50,000. In both network settings showcased in Figure 3, namely (i) $M = 6$ agents, $N = 3$ bands, and jammer in band 3; and (ii) $M = 7$ agents, $N = 5$ bands, and jammer in band 5, agents maintain high throughput and fairness index before, during, and after the jamming episode. We believe that **NCR's inherent event-driven behavior—derived from its spiking foundation— leads to a fast reorganization of the agents when the jammer enters or leaves the network**. Since jamming can be framed as a discrete event, *i.e.*, last band being available (or not) at a given time slot, it seems to naturally fit the event-driving, spiking characteristic of NCR, as evidenced by the high performance obtained in our experiments.

Lastly, we perform an ablation study on the surrogate gradient function and the number of hidden layers of the SNN. This analysis is motivated by the observation that such features are crucial for model performance (both fairness and total throughput). In this study, we consider three possible surrogate gradient functions—arctan, sigmoid, and fast sigmoid—as well as three potential numbers of hidden layers (HL) as 1, 5, and 10. We select $(M, N) \in \{(7, 3), (9, 2), (10, 10)\}$ as experiments to run the ablation study. Table 3 contains the values of the fairness index corresponding to each experiment.

As illustrated in Table 3, when the number of agents and bands is similar ($M \approx N$), fairness index becomes saturated, *i.e.*, both surrogate gradient function and network depth have negligible effects on fairness—it remains greater than 0.999 across the ablation study experiments. On the other hand, scarce-spectrum regimes ($M >> N$) are such that the arctan surrogate gradient function benefits

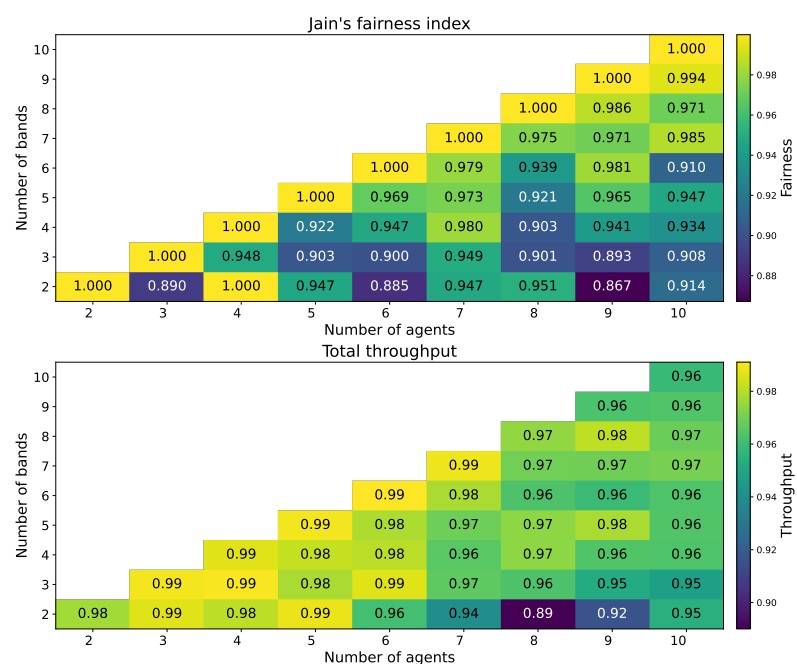

Figure 2: Jain's fairness index and total throughput across all 45 experiments conducted using NCR.

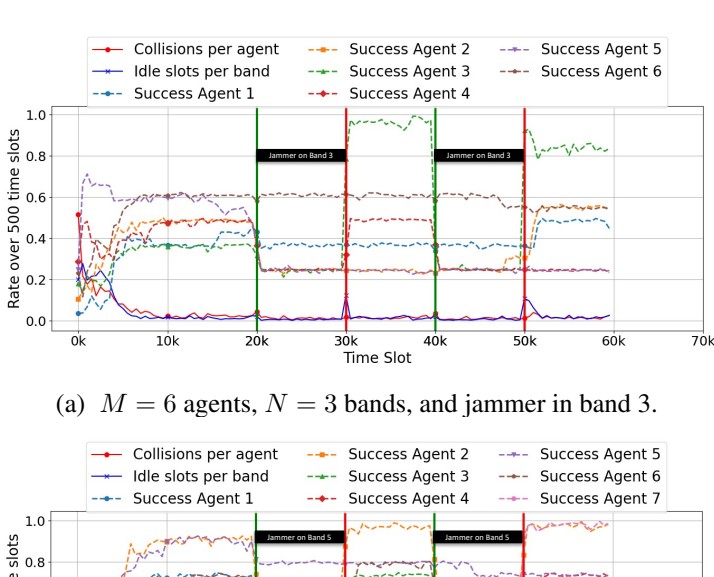

(a) $M = 6$ agents, $N = 3$ bands, and jammer in band 3.

(b) $M = 7$ agents, $N = 5$ bands, and jammer in band 5.

Figure 3: NCR's idle rate, collision rate, and per-agent throughput over time $t$ in a jamming environment. (a) $(M, N) = (6, 3)$ experiment. (b) $(M, N) = (7, 5)$ experiment.

Table 3: Ablation study based on surrogate gradient functions and network depth (HL) for three network settings. The reported values correspond to the Jain's fairness index obtained at each experiment.

|  | Arctan | Sigmoid | Fast Sigmoid |
|---|---|---|---|
| M = 7, N = 3 |  |  |  |
| 1 HL | 0.9494 | 0.9566 | 0.9566 |
| 5 HL | 0.9328 | 0.8164 | 0.9465 |
| 10 HL | 0.9844 | 0.9454 | 0.9092 |
| M = 9, N = 2 |  |  |  |
| 1 HL | 0.8672 | 0.9396 | 0.9052 |
| 5 HL | 0.8730 | 0.8089 | 0.8812 |
| 10 HL | 0.9484 | 0.8550 | 0.8178 |
| M = 10, N = 10 |  |  |  |
| 1 HL | 0.9998 | 0.9995 | 0.9995 |
| 5 HL | 0.9999 | 0.9999 | 0.9999 |
| 10 HL | 0.9999 | 0.9999 | 0.9998 |

from increased network depth. As for alternative surrogate gradient functions (fast sigmoid and sigmoid), more hidden layers do not reliably improve fairness index levels. Such results suggest that, overall, when bands are scarce relative to agents, arctan with deeper SNNs leads to higher fairness index results. Otherwise, shallow SNNs can achieve competitive fairness regardless of the chosen surrogate gradient function. Moreover, all experiments in the ablation study have achieved throughput greater than 85%—with the vast majority of them even greater than 93%.

## 5 CONCLUSION

In this paper, we introduce the Neuro-Cognitive Radio, a novel neuromorphic-based approach that addresses challenging DSA scenarios by leveraging spiking neural networks to enable efficient, fair, and decentralized spectrum sharing without requiring centralized training or explicit information exchange among agents. Our SNN backbone—built on leaky-integrate-and-fire neurons—incorporates temporal dependencies through a sliding window of past actions and outcomes, achieving superior performance in fairness while maintaining high throughput across diverse network configurations.

The simulation results show that NCR consistently outperforms the baseline DQN method in terms of fairness. For example, Jain's fairness index improves from 0.8041 to 0.9732 in the scenario with 7 agents and 5 bands, while maintaining competitive throughput levels. This performance is evident in the convergence patterns of agent success rates, where NCR promotes equitable access in congested environments and even under jamming conditions. Moreover, as suggested by the ablation analysis, the optimal choice of SNN depth and surrogate gradient function strongly depends on the network settings: while scarce-spectrum experiments ($M >> N$) seem to benefit from more hidden layers and arctangent, other cases ($M \approx N$) achieve high fairness regardless of the choices we make.

These results underscore the potential of neuromorphic computing for DSA, capitalizing on its event-driven, parallel processing to achieve high fairness and total throughput. As the main focus of this work, NCR has been applied to several experiments, ensuring robust and reliable results in a simulated environment. Looking ahead, we will use these insights as a foundational step toward deploying spiking models on real neuromorphic hardware platforms (Intel Corporation, 2025)—leading to tangible energy efficiency compared to traditional von Neumann architectures. Future work could explore scaling to larger networks (*e.g.*, hundreds of agents), integrating real-world channel impairments like fading or interference, and hybrid approaches combining SNNs with other RL variants for enhanced robustness. Additionally, empirical validation on physical testbeds would bridge the gap between simulation and practical implementation, paving the way for neuromorphic solutions in next-generation, energy-efficient wireless systems.

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
