# OpenReview forum: "Neuro-Cognitive Radios for Dynamic Spectrum Access"
_ICLR.cc/2026/Conference — ICLR 2026 Conference Withdrawn Submission_

### Official Review · Reviewer_EoZK · 2025-10-29

**Soundness:** 3
**Presentation:** 3
**Contribution:** 3
**Rating:** 4
**Confidence:** 4

**Summary:**

This paper introduces Neuro-Cognitive Radios (NCR) , a novel framework that applies Spiking Neural Networks (SNNs) combined with reinforcement learning to solve the Dynamic Spectrum Access (DSA) problem in wireless communications.

Key Contributions
1.First integration of SNNs for DSA.(Maybe during the work because similar method has been found to deal with the problem of DSA)
2.Achieves significantly higher fairness than traditional DQN while maintaining similar throughput.
3.Handles dynamic environments with time-varying spectrum availability.
4.Demonstrates the potential of neuromorphic computing for energy-efficient, adaptive wireless networking.

**Strengths:**

Originality: The paper introduces a novel combination of spiking neural networks (SNNs) and decentralized dynamic spectrum access (DSA) to address fairness in wireless networks. This creative integration of neuroscience-inspired architectures (SNNs) with wireless resource allocation represents a fresh perspective in the field. While SNNs have been explored in reinforcement learning (RL), their application to fairness-centric decentralized DSA problems is relatively uncharted, offering a unique angle for solving resource allocation challenges in dynamic environments.
Quality: The experimental design demonstrates rigor in several aspects, including comprehensive evaluation,  ablation analysis and visual validation.
Clarity: The paper is generally well-structured, with a clear problem formulation (Section 2) and logical flow from methodology to results. The use of standardized metrics (Jain’s fairness index, throughput) ensures interpretability.
Significance: The paper addresses a critical challenge in wireless networks: achieving fairness in decentralized DSA without sacrificing throughput. This problem is increasingly relevant for 5G/6G and IoT applications, where autonomous spectrum sharing among devices is essential. The framework’s demonstrated ability to maintain fairness under interference (Figure 3) adds practical value.

**Weaknesses:**

Lack of Comparative Baselines with State-of-the-Art RL Algorithms: The paper compares NCR primarily with DQN, a relatively basic RL algorithm, without benchmarking against more advanced decentralized multi-agent RL methods.
Insufficient Theoretical Justification for Fairness Improvement: The paper attributes fairness improvements solely to SNN's "event-driven" nature without providing a theoretical explanation for why this architecture leads to better fairness. For example, it does not analyze how the spiking dynamics affect the decision-making process in terms of resource allocation fairness.
Insufficient Detail on Spectrum Access Environment Setup: The paper lacks critical details about the spectrum access environment, which is essential for understanding the experimental context and evaluating the results. Specifically, they include spectrum characteristics, channel model, network topology and interference model.

**Questions:**

Q1: Why was DQN chosen as the primary baseline instead of more advanced decentralized multi-agent RL algorithms ? Could the authors include comparisons with at least one modern (decentralized) RL algorithms to validate NCR’s claimed advantages over existing methods?
Q2: The paper attributes fairness improvements to SNN’s "event-driven" nature but provides no theoretical analysis. Could the authors formalize how spiking dynamics (e.g., leaky integrate-and-fire neuron behavior) mathematically influence fairness metrics like Jain’s index?
Q3: Table 3 shows ablation studies for fairness but not throughput or convergence speed. Could the authors analyze how SNN hyperparameters (e.g., leakage rate β, threshold θ) affect trade-offs between fairness, throughput, and training stability?

---

### Official Review · Reviewer_B3QC · 2025-10-30

**Soundness:** 2
**Presentation:** 2
**Contribution:** 2
**Rating:** 2
**Confidence:** 3

**Summary:**

This paper proposes Neuro-Cognitive Radio (NCR), a neuromorphic-based framework integrating Spiking Neural Networks (SNNs) and Reinforcement Learning (RL) for decentralized Dynamic Spectrum Access (DSA). NCR aims to enhance fairness among wireless devices while maintaining high throughput, addressing limitations of existing RL-based solutions (e.g., DQN) that lack fairness or require complex coordination. Simulations across 50+ network settings (static, time-varying with jammers) show NCR outperforms DQN in Jain’s fairness index (up to 2.5x improvement) with comparable throughput. Ablation studies explore impacts of SNN depth and surrogate gradient functions on performance.

**Strengths:**

1. This manuscript shows strong empirical validation where extensive experiments (varying agents/bands, jamming) and direct comparison with DQN demonstrate robust fairness-throughput tradeoff.

2. Fully decentralized training/execution, no inter-agent communication, suitable for resource-constrained radios.

**Weaknesses:**

1. Energy efficiency, a key claimed advantage of neuromorphic computing, is not quantified, but is only hypothesized for future work.

2. Lack of real-world hardware validation; simulations rely on idealized channel models (no fading/interference impairments).

3. Scalability to large-scale networks (hundreds of agents) is untested, limiting generalizability to beyond-5G use cases.

4. The reward function’s design (e.g., weights for recent transmissions) is not justified with comparative analysis (e.g., vs. simpler reward schemes).

**Questions:**

Please refer to the weakness part

---

### Official Review · Reviewer_Mdfy · 2025-11-01

**Soundness:** 3
**Presentation:** 3
**Contribution:** 2
**Rating:** 0
**Confidence:** 5

**Summary:**

The goal of this work is to improve wireless spectrum access by leveraging neuromorphic computing. The authors design and implement this approach, and assess its performance.

**Strengths:**

+ The idea of using neuromorphic computing for dynamic spectrum access is useful.
+ Experiments are extensive and meaningful.

**Weaknesses:**

- The relevance of this work to ICLR is quite tenuous. This work focuses primarily on the wireless networking piece and not on machine learning or representation learning. This is more appropriate for venues like IEEE INFOCOM or IEEE ICC.
- The authors do not discuss the complexity associated with deploying this architecture in a real wireless system.
- While dynamic spectrum access is a meaningful problem, recent advances in wireless networks allow performing sensing effectively with databases and/or simple learning rules. It is not clear why such a neuromorphic approach is useful.
- Despite being extensive, the experiments are unclear on the type of networks (e.g., cellular? WiFi?) and there are thin details on the wireless network parameters adopted.

**Questions:**

- What are the core machine learning advances provided by this work that justify its relevance to ICLR?
- How many devices can your approach handle? How does it scale with hundreds of devices?
- Do you take into account interference and fading in your models? How are those impacting your solution?
- What happens if you have a learning error? Does the network transmit and cause additional interference?
- How does your approach compare to simpler solutions already adopted in the wireless literature?

---

### Official Review · Reviewer_r1fk · 2025-11-03

**Soundness:** 2
**Presentation:** 2
**Contribution:** 1
**Rating:** 0
**Confidence:** 5

**Summary:**

The paper tried to use neuromorphic computing to solve dynamic spectrum access (DSA) problem.

**Strengths:**

It is interesting to apply machine learning approaches to address DSA issues.

**Weaknesses:**

The paper lacks basic understanding of dynamic spectrum access (DSA) or even wireless communications. The wireless model for DSA is oversimplified, which ignores the nature of DSA, i.e., frequency reuse. Besides, it is not clear what is the underlying wireless network architecture: is it cellular network, ad hoc network, mesh network, device-to-device communications, satellite networks, etc.? Is it 3G, 4G, LTE, 5G, wifi, bluetooth, etc. What is the PHY layer technique we are discussing: Is it TDMA, FDMA, OFDM, MIMO, etc.? Many MAC/networking issues are not discussed at all, e.g., hidden terminal problems in wireless transmissions.

While machine learning may be a good tool for a lot of challenging problems in many areas, it is worth a while to learning some basic domain knowledge before directly hammering a naive machine learning scheme into a well-studied domain.

**Questions:**

- Why the nature of DSA, i.e., frequency reuse, is not considered?
- What is the underlying wireless network architecture: cellular network, ad hoc network, mesh network, device-to-device communications, satellite networks, etc.?
- Is it 3G, 4G, LTE, 5G, wifi, bluetooth, etc.?
- What is the PHY layer technique we are discussing: Is it TDMA, FDMA, OFDM, MIMO, etc.?
- What kinds of interferences are discussed in this paper? Does it use PHY model or protocol model?
- How to address hidden terminal issues?

---

### Note · Authors · 2026-01-30

I have read and agree with the venue's withdrawal policy on behalf of myself and my co-authors.

---

### Meta-Review · Area_Chair_W9Ks · 2026-01-06

**Summary:**

The submission proposes Neuro-Cognitive Radios (NCR) that combine SNNs + RL for decentralized dynamic spectrum access, reporting improved fairness over a DQN baseline while maintaining competitive throughput under several simulated settings (including jamming). Across reviews, the primary concerns driving the decision are: (i) oversimplified / unclear wireless system modeling (network type, PHY/MAC assumptions, interference/fading, frequency reuse, hidden terminals), (ii) limited ML contribution and tenuous fit to ICLR, and (iii) insufficient baselines and realism (e.g., only DQN; no hardware or energy-efficiency quantification; unclear scalability).

**Reviewer Concerns:**

No rebuttal/discussion content was available in the provided forum snapshot; thus, no concerns can be verified as resolved.

**Reviewer Scores:**

Given the (apparent) absence of rebuttal/discussion and the strength of the core concerns, I expect little rating change.

---

### Decision · Program_Chairs · 2026-01-26

Reject